# Directed Functional Coordination Analysis of Swallowing Muscles in Healthy and Dysphagic Subjects by Surface Electromyography

**DOI:** 10.3390/s22124513

**Published:** 2022-06-15

**Authors:** Yiyao Ye-Lin, Gema Prats-Boluda, Marina Galiano-Botella, Sebastian Roldan-Vasco, Andres Orozco-Duque, Javier Garcia-Casado

**Affiliations:** 1Centro de Investigación e Innovación en Bioingeniería, Universitat Politècnica de València, 46022 Valencia, Spain; yiye@ci2b.upv.es (Y.Y.-L.); marinagaliano27@gmail.com (M.G.-B.); jgarciac@ci2b.upv.es (J.G.-C.); 2Grupo de Investigación en Materiales Avanzados y Energía, Instituto Tecnológico Metropolitano, Medellin 050034, Colombia; sebastianroldan@itm.edu.co; 3Grupo de Investigación e Innovación Biomédica, Instituto Tecnológico Metropolitano, Medellin 050034, Colombia; andresorozco@itm.edu.co

**Keywords:** surface electromyography, Granger causality, functional coordination, dysphagia, swallowing muscle coupling

## Abstract

Swallowing is a complex sequence of highly regulated and coordinated skeletal and smooth muscle activity. Previous studies have attempted to determine the temporal relationship between the muscles to establish the activation sequence pattern, assessing functional muscle coordination with cross-correlation or coherence, which is seriously impaired by volume conduction. In the present work, we used conditional Granger causality from surface electromyography signals to analyse the directed functional coordination between different swallowing muscles in both healthy and dysphagic subjects ingesting saliva, water, and yoghurt boluses. In healthy individuals, both bilateral and ipsilateral muscles showed higher coupling strength than contralateral muscles. We also found a dominant downward direction in ipsilateral supra and infrahyoid muscles. In dysphagic subjects, we found a significantly higher right-to-left infrahyoid, right ipsilateral infra-to-suprahyoid, and left ipsilateral supra-to-infrahyoid interactions, in addition to significant differences in the left ipsilateral muscles between bolus types. Our results suggest that the functional coordination analysis of swallowing muscles contains relevant information on the swallowing process and possible dysfunctions associated with dysphagia, indicating that it could potentially be used to assess the progress of the disease or the effectiveness of rehabilitation therapies.

## 1. Introduction

A complete swallow requires a complex sequence of highly regulated and coordinated oral and pharyngeal events for the safe passage of a bolus into the oesophagus without compromising the airway [1]. Dysphagia is an inability to swallow foods or liquids properly that affects 9.4 million adults every year (1 in 25) in the United States [2]. The prevalence of dysphagia ranges between 12–13% in hospitalized patients, but rises to 30% in the elderly and to 60% in intensive care unit and home-care patients [3,4,5,6]. Dysphagia is usually caused by another condition, such as aging, neurological diseases, neuromuscular impairment, head and neck cancers, and gastro-oesophageal reflux disease, among others. Dysphagia can produce malnutrition, dehydration, aspiration, pneumonia, and even death, as well as a reduced quality of life, social isolation, and healthcare-related costs [6,7,8]. The mean attributable cost of dysphagia is about USD 12,715, 40.36% more than nondysphagic hospitalized patients [9]. Patel estimated that dysphagia was responsible for between USD 4.3 to 7.1 billion in additional hospital costs annually in the US [8].

A videofluoroscopic swallowing study is the reference diagnostic method for assessing oropharyngeal dysphagia [10]. This technique involves patient exposure to ionizing radiation [10], so it is not recommended for patient follow-up when evaluating the effectiveness of rehabilitation, although it does not always identify neuromuscular abnormalities in pharyngeal or laryngeal physiology [10]. An example of the latter could be patients with muscle tension dysphagia who present functional dysphagia but exhibit normal oropharyngeal and oesophageal swallowing function, as evidenced by videofluoroscopic swallow study [11,12]. Surface electromyography (sEMG) has emerged as a simple, nonradioactive, and noninvasive method of measuring the patterns of muscle activity during swallowing and allows clinicians to describe the physiology of swallowing behaviour [13]. sEMG is the extracellular recording of the electrical activity of muscle fibres on the skin surface, which reflects the electrophysiological muscle response to nerve stimulation. Most swallowing-related studies have used sparse sEMG electrodes to sense the activity and have extracted electrophysiological information from face and neck muscles, e.g., from the masseteric, submental (suprahyoid), and infrahyoid regions [14,15,16,17,18,19,20]. Unlike classical temporal and spectral parameters from a single sEMG channel, sEMG signal characterization cannot precisely characterize the complete swallowing process.

A complete swallowing process in an adult human is an autonomous motor behaviour requiring not only the coordination of 26 muscles and 5 cranial nerves, but also coordination with mastication and respiration [21]. According to Bernstein’s motor control theory and coordination dynamics theory, motor behaviour involves the coupling of different physiologic structures, such as muscles, in task-specific control units known as *synergies* [22]. In most motor control synergies, a common neurologic activation pattern associated with a specific task objective is responsible for providing temporary flexible couplings between muscle systems [22]. As a high-level neuromuscular synergy driven by the skeletal muscles and smooth muscles of the pharynx and oesophagus is required in swallowing to successfully execute the swallowing sequences [23,24], the coordination of different swallowing muscles, also known as *muscle coupling*, exhibits the hallmark characteristics of motor control synergy. This coordination is likely to be altered in dysphagic subjects, and its analysis can, thus, provide new indicators for its early detection or for the quantitative evaluation of rehabilitation therapies.

Several previous works have studied the coordination of swallowing muscles using different approaches. Most have analysed the activation sequences and coordination patterns of the different muscles involved in swallowing using sparse electrode pairs: suprahyoid [25]; oral and laryngeal [26]; and laryngeal, pharyngeal, and submental [16,27]. The analysis of sequence activation has usually been performed by detecting the fiducial point of muscle activation, such as the onset and offset timing, from rectified and integrated sEMGs [15,17,27] to determine the temporal relationship between the muscles and to establish the activation sequence pattern. Using the sEMGs from 15 electrodes in the face and throat, McKeown et al. successfully detected the swallowing pattern by separating laryngeal excursion, tongue movement, and activation of the buccal and masseteric musculature using independent component analysis [28]. Zhu et al. showed the feasibility of obtaining sequential sEMG energy maps from a 2D high-density electrode array on the submental and infrahyoid muscles to analyse spatiotemporal properties during swallowing [1]. They found that the sEMG potential maps constructed from a sliding window mainly reflected the submental and infrahyoid muscles associated with the high intensity on the top and in the centre of the maps, respectively [1]. Dysphagic subjects were shown to present a significantly higher prevalence of inverted muscle activation patterns, i.e., infrahyoid-related was followed by suprahyoid-related activation [15]. 

Few studies in the literature have been conducted to assess the functional relationship or coordination between swallowing muscles with sEMG. Our previous work assessed the crosstalk and synchronization properties of sEMGs from swallowing muscles based on cross-correlation [14]. We found the correlation of bilateral suprahyoid muscles to be moderate and slightly higher than that of bilateral infrahyoid muscles [14]. Wang et al. proposed a discoordination index based on cross-correlation coefficients that reflected the differences between the surface EMG patterns of bilateral muscle groups [13]. Lee et al. found that suprahyoid muscle activity showed a significant positive correlation with infrahyoid activity [29]. Covariance was also proposed to examine patterns of functional independence for tongue muscles during speech and swallowing [30]. Steele et al. found that sequential liquid swallowing was associated with increased frequency entrainment through cross-spectral coherence analysis and reduced relative phase variability in tongue–jaw coordination [22]. The EMG–EMG transfer function and coherence function in the first peak frequency were used to assess jaw and neck muscle coordination in rhythmic chewing [31]. The authors found that the coherence and phase in nonchewing, side-neck muscle activities exhibited a significant negative correlation [31].

The correlation coefficient, or lagged correlation, measures the linear relationship in a time domain between two time series [32]. Frequency entrainment [22] and transfer function [31] estimates from cross-spectra or coherence are traditional measures of linear correlations in the frequency domain [32]. These measures are seriously affected by instantaneous interactions or coupling associated with volume conduction, as it is challenging to differentiate them from true interactions [32]. They provide information only on the interchannel interaction strength, not on the directionality of the interaction [32], which is a relevant physiological characteristic of swallowing. Granger causality (G-causality) has been widely used to determine brain functional connectivity to identify regional activations and to characterize functional circuits from functional magnetic resonance imaging, electroencephalography, and magnetoencephalography [33,34,35,36]. Based on the hypothesis that causes precede and help to predict effects and that manipulations of the cause change the effects [37], G-causality provides a statistical measurement of functional interaction strength based on the relative prediction improvement to identify linear directional interdependence between multivariate time series [32,36]. This is a data-driven approach that estimates the causal statistical influences without the need for physical intervention that is able to quantify the directional flow of information [33]. The analysis of muscular interactions, including both strength and directionality, can potentially provide a better understanding of the underlying physiology of swallowing and the possible alterations in dysphagic subjects.

This work thus aims to assess functional muscular interactions during swallowing in healthy individuals and the possible changes in dysphagic subjects by analysing the G-causality of their sEMGs. For this, we characterize and compare both the directionality and the coupling strength of multiple swallowing muscles with different bolus consistencies.

## 2. Materials and Methods

### 2.1. Data Acquisition

The database was composed of sEMG recordings from 30 healthy volunteers (17 males and 13 females with average ages of 42.2 ± 15.5 and 46.5 ± 17.6 years old, respectively) and 31 subjects with dysphagia (19 males and 12 females with average ages of 42.4 ± 17.1 and 48.7 ± 13.2 years old, respectively). The volunteers signed an informed consent approved by the Ethics Committee of the Instituto Tecnológico Metropolitano (Medellín, Colombia). All the patients included in the study had confirmed diagnoses of functional oropharyngeal dysphagia due to neurogenic causes (multiple sclerosis: 7; amyotrophic lateral sclerosis: 6; cerebral palsy: 4; muscular dystrophy: 4; ischemic stroke: 3; Parkinson’s disease: 3; traumatic brain injury: 3; and secondary hydrocephalus (neurocysticercosis: 1)).

To reduce skin-electrode impedance, we first shaved the skin under the electrodes as required and then exfoliated with abrasion gel (Nuprep, Weaver and Company, Aurora, CO, USA) before cleaning the recording surface with isopropyl alcohol. Bipolar sEMG activity was recorded from the following three bilateral muscle groups involved in the oral and pharyngeal swallowing phases using 6 pairs of disposable Ag/AgCl electrodes (3M electrodes Ref. 2228; inter-electrode distance of 25 mm): the left and right masseters (LM and RM), the left and right suprahyoid (LSH and RSH), and the left and right infrahyoid (LIH and RIH). Figure 1 shows the electrode placement for multichannel sEMG acquisition. A Noraxon Ultium EMG amplifier (Noraxon USA; CMMR > 100 dB, 16 bits A/D converter) was used for signal collection and conditioning. The bipolar sEMG signals were amplified and band-pass filtered within 10 and 500 Hz and were sampled at 2 kHz.

The recording protocol consisted of three swallowing tasks commonly used in dysphagia screening [38,39]: saliva, 10 mL of water, and 10 mL of liquid yoghurt. A 1.5 oz cup was used to deliver water and yoghurt to the oral cavity. The subjects were seated in an upright position and asked to swallow the bolus as naturally as possible.

### 2.2. Conditional Granger Causality

Conditional Granger causality (cG-causality) is defined as the degree to which the past of target Y helps to predict source X beyond the degree to which X is already predicted by its own past and the past of the conditioning variable Z [40]. cG-causality can be used to detect real interactions between different muscles, avoiding false causality due to their underlying ‘hidden’ interactions [40]. Based on vector autoregressive model theory, we considered the full (see Equation (1)) and reduced (Equation (2)) regressions of *X,* including the conditioning variable *Z* in both regressions:(1)$$X_{t}=\sum_{k=1}^{p}A_{XX,k}\cdot X_{t-k}+\sum_{k=1}^{p}A_{XY,k}\cdot Y_{t-k}+\sum_{k=1}^{p}A_{XZ,k}\cdot Z_{t-k}+\varepsilon_{X,t}$$
(2)$$X_{t}=\sum_{k=1}^{p}{A^{\prime }}_{XX,k}\cdot X_{t-k}+\sum_{k=1}^{p}{A^{\prime }}_{XZ,k}\cdot Z_{t-k}+{\varepsilon^{\prime }}_{X,t}$$
where *p* is the vector autoregressive model order, the coefficients *A_XX,k_* represent the autodependence of *X* on its own past, and *A_XY,k_* and *A_XZ,k_* are the coefficients of the dependence of *X* on the past of *Y* and on the past of the conditioning variable *Z,* respectively. $\varepsilon_{X,t}$ are the model regression residuals with the covariance matrix $\Sigma_{XX}\equiv \mathrm{cov}\left(\varepsilon_{X,t}\right)$.${A^{\prime }}_{XX,k}$, ${A^{\prime }}_{XZ,k}$ are the corresponding reduced regression coefficients, and ${\varepsilon^{\prime }}_{X,t}$ are the residual covariance matrices of the reduced regression ${\Sigma^{\prime }}_{XX}\equiv \mathrm{cov}\left({\varepsilon^{\prime }}_{X,t}\right)$.

cG-causality from *Y* to *X* $F_{Y\to X|Z}$ is, thus, defined as the log-likelihood ratio of the determinant of the residual covariance matrix taking into account the joint effects of Z (see Equation (3)). cG-causality attempts to quantify the degree to which the full regression represents a “better” model of the data than the reduced regression.
(3)$$F_{Y\to X|Z}\equiv \frac{\left|{\Sigma^{\prime }}_{XX}\right|}{\left|\Sigma_{XX}\right|}$$

We used pairwise-conditional G-causality [40] of the multichannel EMG data ${G\;}_{i,j}\left(EMG\right)$, which determines the causality *EMG_j_*–*EMG_i_* (from muscle *j* to *i*) conditioning out all the known remaining data (see Equation (4)):(4)$${G\;}_{i,j}\left(EMG\right)\equiv F_{EMG_{j}\to EMG_{i}|EMG_{\left[ij\right]}}$$
where the subscript [_ij_] denotes omission of the *i_th_* and *j_th_* variables in the multivariate EMG data. Details of the algorithm can be found in the original work by Barnett [40].

The optimal order of the vector autoregresssive model to achieve a compromise between the model’s precision and complexity was determined with the Akaike information criterion [40].

### 2.3. Data Analysis

As mentioned above, for a number of reasons we preferred to analyse the interactions between different swallowing muscles with pairwise cG-causality using sliding windows of a fixed length rather than the whole recording or contraction epochs. Firstly, there is no reliable tool for the automatic identification of the onset and offset of the contraction epoch in sEMG, which remains a challenge for the scientific technical community. Secondly, swallowing EMG data may present a nonstationary nature, such as EMGs from other muscles [41], which fails in the assumption of cG-causality on the covariance stationary stochastic process [40]. Both the contraction epoch and the whole EMG recording have a variable duration, which could influence the Granger causal inference analysis [42]. The sliding window length was set to 0.5 s with no overlap based on the timing values of the pharyngeal swallowing phase [38,43]. This selection was experimentally confirmed through comparison with a 1s window, for which the average percentage (see details below) of the cG-causality value did not show consistent trends between healthy and dysphagic subjects.

For each subject, bolus, and muscle pair, we obtained an array of cG-causality values as the results of the sliding window. Since swallowing muscles are expected to present higher interactions during contraction than in basal activity (at rest), we considered the maximum cG-causality value of each interaction pair as their interaction during swallowing-muscle contraction, obtaining maximum interaction matrix—MIM—(see Figure 2), so that one 6×6 MIM was obtained per subject in which each row and column represented one acquisition channel.

We then sought to determine the relevant interaction pairs for each subject by discarding interactions with weak coupling strengths, i.e., below a threshold of 25% of the maximum value of each subject’s MIM (MIM_max_) or the threshold interaction matrix. For each interaction pair, we evaluated the percentage of subjects that exceeded this threshold for each bolus. The average percentage of the three boluses (average percentage matrix) was then used to compare the occurrence of a relevant interaction between different pairs of muscle groups, after which the mean value of the average percentage matrix was computed for the bilateral, ipsilateral, and contralateral muscles of healthy and dysphagic subjects. For the value of the threshold used, we aimed to achieve a trade-off between the sensitivity of detecting relevant interactions by a physiological interpretation and obtaining ‘spurious’ interactions. True interactions associated with swallowing should be reproducible in almost all healthy subjects, achieving approximately 100% of the subjects that exceeded this threshold. In addition, some of these interactions may be altered in dysphagic subjects. Specifically, in this work, we compared different threshold values, ranging from 10–40%, and obtained similar results for threshold ranges of from 20 to 30%.

The cG-causality difference between healthy and dysphagic subjects was assessed using the raw MIM, which represented muscle interaction during swallowing. Firstly, cG-causality values of healthy and dysphagic subjects were compared for each bolus type and muscle pair using the Wilcoxon rank-sum statistical test (α = 0.05). Secondly, we analysed the bolus effect on cG-causality for each interaction pair using the Friedman test (α = 0.05) for both types of subjects. Finally, we evaluated the interaction asymmetry of each muscle pair (from *i* to *j* muscles vs. from *j* to *i* muscles) using the Wilcoxon signed-rank test (α = 0.05) for both subject types.

## 3. Results

### 3.1. Assessment of Muscle Interactions

Table 1 shows the average percentage of subjects that exceeded the pre-established threshold of MIM_max_ for the three boluses with each muscle-pair interaction. Bilateral suprahyoid (RSH⟷LSH) and infrahyoid (RIH⟷LIH) muscles, in general, reacted closely with each other, with high occurrence between patients. The percentage of subjects who exceeded the pre-established threshold was greater than 90%, while the bilateral masseter muscle interaction (RM⟷LM) was relevant in a smaller percentage of subjects (70–80%). As expected, regardless of the muscle groups, the bilateral muscles in healthy subjects seemed to activate simultaneously, retrieving a similar average percentage in both directions. Dysphagic subjects obtained a similar average percentage for bilateral muscle interactions. We also found consistent interactions among subjects for ipsilateral suprahyoid-infrahyoid muscles (RSH⟷RIH and LSH⟷LIH), with 88% of the healthy subjects exceeding the threshold. The ipsilateral couplings between masseter and suprahyoid (RM⟷RSH and LM⟷LSH) or infrahyoid (RM⟷RIH and LM⟷LIH) muscles were moderately consistent between healthy subjects, with over 76% of the individuals exceeding the threshold. In dysphagic subjects, the consistency of the average ipsilateral interaction was considerably reduced (87% healthy vs. 78% dysphagia). The greatest differences between healthy and dysphagic subjects occurred in the left ipsilateral interactions: LSH→LM (94% vs. 69%) and LIH→LM (90% vs. 72%). We found that the contralateral interaction was relatively weaker (68% healthy vs. 64% dysphagia). As the bilateral and ipsilateral muscles presented higher consistencies of relevant interactions than contralateral muscles in both healthy and dysphagic subjects, in the succeeding sections we, therefore, focus on the bilateral interactions and ipsilateral masseter–suprahyoid and suprahyoid–infrahyoid interactions due to their sequential activations under physiological conditions.

Figure 3 shows the cG-causality for bilateral muscles in healthy and dysphagic subjects for the three boluses. Next, we describe the difference between healthy and dysphagic subjects, the effect of bolus consistency, and the interaction symmetry for each pair of muscles. The results of the statistical analysis are described in Section 3.2.

*Bilateral Masseter.* Dysphagic subjects showed increased cG-causality in bilateral masseters for saliva and water boluses when compared to healthy subjects with little difference in yoghurt. We found a higher coupling strength of the RM→LM interaction for water and yoghurt than for saliva in the healthy group, but not in dysphagic subjects. The bilateral masseter cG-causality in both directions was similar for both healthy and dysphagic subjects, except for with saliva in the first group.

*Bilateral suprahyoid.* These muscle interactions tended to decrease more in dysphagic than in healthy subjects. We did not find a clear bolus effect of cG-causality in healthy subjects, while it tended to be higher for water and yoghurt than for saliva in dysphagic subjects. Again, bilateral suprahyoid cG-causality was similar in both directions, except for yoghurt in healthy subjects.

*Bilateral infrahyoid.* Dysphagic subjects, in general, obtained higher medians of cG-causality than healthy subjects. We found higher medians of cG-causality in the RIH→LIH interaction for the water and yoghurt boluses than for the saliva bolus in healthy subjects. The LIH→RIH interaction showed higher cG-causality for water and yoghurt than for saliva in both subject groups, being more evident in dysphagics. Bilateral infrahyoid muscles showed similar cG-causality in both directions in healthy subjects, while dysphagic subjects had an asymmetric interaction.

Figure 4 gives the cG-causality of ipsilateral muscles in healthy and dysphagic subjects for the three boluses. The results obtained from each muscle pair is again described, differentiating between healthy and dysphagic subjects, bolus consistencies, and the interaction symmetry.

*Ipsilateral masseter**↔**suprahyoid.* Dysphagic subjects usually obtained slightly higher than or similar cG-causalities to healthy individuals, with no clear trend. We also found an upward trend in cG-causality from saliva to yoghurt for healthy subjects, while this trend was only apparent in LM→LSH for dysphagic subjects. In healthy subjects, the masseter⟷suprahyoid muscle interaction seemed to be symmetric. In general, we found an asymmetric masseter→suprahyoid interaction, except for with the yoghurt bolus on the right side in dysphagic subjects, while the upward direction tended to achieve greater cG-causality.

*Ipsilateral suprahyoid*⟷*infrahyoid*. Dysphagic subjects generally obtained similar or slightly lower cG-causalities in the downward direction than healthy subjects, while dysphagic subjects tended to obtain higher cG-causality values than healthy subjects in the upward direction, especially in RIH→RSH for both saliva and yoghurt. We found an upward trend in cG-causality from saliva to yoghurt in healthy subjects, except for RIH→RSH, but not in dysphagic subjects. We also found a predominantly downward interaction for both left and right suprahyoid→infrahyoid muscles in healthy subjects, which matched with the direction of swallowing. In dysphagic subjects, this predominance was notably only observed in the LSH→LIH interaction.

### 3.2. Statistical Analysis

Figure 5, Figure 6 and Figure 7 illustrate the muscle interactions with statistically significant differences between healthy and dysphagic subjects, between boluses, and between directions, respectively. Statistically significant differences were found between healthy and dysphagic subjects for the RIH→LIH bilateral interaction for saliva (Figure 5A), as well as for the ipsilateral interactions of RIH→RSH and LSH→LIH, also for saliva (Figure 5A), and for RIH→RSH for yoghurt (Figure 5C). Despite weaker contralateral vertical interactions, we found statistically significant differences for RIH→LM in water and yoghurt and for ⟷ in yoghurt. The greatest significant differences between healthy and dysphagic subjects were, notably, obtained with yoghurt.

No significant differences were found between the boluses for healthy subjects (Figure 6A). In dysphagic subjects, statistically significant differences between the boluses were found in the bilateral infrahyoid interaction LIH→RIH, the left ipsilateral vertical interactions of LM→LSH and LSH→LIH, and the contralateral vertical interaction of RIH→LM (Figure 6B).

For muscle interaction directionality, we found significant asymmetric interactions with a downward-dominant direction in healthy subjects for the RSH⟷RIH interaction for yoghurt and in the LSH⟷LIH interactions for water and yoghurt (Figure 7B,C). In dysphagic subjects, the asymmetric muscle interactions only appeared in the bilateral infrahyoid muscle interaction and ipsilateral LSH⟷LIH for saliva (Figure 7A).

## 4. Discussion

### 4.1. Relevant Muscle Interactions during Swallowing

In this work, we evaluated the functional coordination of three muscle groups involved in swallowing, i.e., masseters, suprahyoid, and infrahyoid. To the best of the authors’ knowledge, this is the first exploratory work to study the directed functional coordination of swallowing muscles by means of cG-causality, which is less-influenced by the volume conduction effect than simple correlation or coherence-based approaches [32,44]. Regardless of subject group and bolus type, we found both bilateral and ipsilateral suprahyoid and infrahyoid muscle activities to be strongly coupled, while the interactions of the bilateral masseter, the ipsilateral masseter, and the suprahyoid muscles were weaker. The high degree of supra- and infrahyoid coupling agrees with other authors who found a significant positive correlation between these muscles [29]. We mostly found a high level of bilateral suprahyoid rather than bilateral infrahyoid coupling in healthy subjects, which agrees with our previous study in assessing the functional coordination by cross-correlation [14]. The lower interactions for bilateral masseter muscles may be associated with their being voluntary movements. Only swallowing tasks that involved more supra- and infrahyoid activation than masseteric were assessed, since no chewing was required. In general, we found a high overall occurrence of relevant interactions (high average percentage, see Table 1) of supra- and infrahyoid muscles under physiological conditions, which is consistent with the electrophysiology of swallowing [45]. We also found a predominantly downward direction in ipsilateral supra- and infrahyoid muscles, which matches with the transit of the physiological-descendent bolus. Under physiological conditions, the suprahyoid were activated 95 ms earlier than the infrahyoid muscles [46]. This may justify the relatively higher cG-causality value of suprahyoid-to-infrahyoid interaction (improved predictability of infrahyoid EMG activity with the known EMG activity from the suprahyoid muscle) than the value for infrahyoid-to-suprahyoid interaction. In addition, we found that the overall occurrence of ipsilateral interactions was considerably reduced in dysphagic subjects. The reduced strength of the swallowing interaction coupling could be associated with the alteration of stereotyped motor behaviours and could also be a potential dysphagia biomarker [47].

### 4.2. Alterations of Muscle Interactions in Dysphagia

We did not find significant differences in functional coordination between the bilateral masseters, ipsilateral masseters, and suprahyoid muscles in either healthy or dysphagic subjects. Our results suggested that the main alterations in dysphagic subjects are mainly found in the supra- and infrahyoid muscles. This finding may be associated with the fact that the primary function of the masseter muscles is to raise the jaw when chewing and to act as mandibular stabilizers during swallowing [48]. In this regard, Zanato et al. showed that swallowing water demanded a greater activation of the suprahyoid than the masseter muscles compared to the values at rest (9.57 μV compared to 3.81 μV and 6.15 μV compared to 3.47 μV, respectively) [49]. Monaco et al. found that masseter muscles showed lower rectified sEMG values than submental groups during spontaneous saliva swallowing, even when activated [50].

Our preliminary results showed a significantly increased interaction in RIH→RSH for both saliva and yoghurt, in RIH→LIH and LSH→LIH for saliva, and in RIH⟷LSH for yoghurt in dysphagic subjects, suggesting an altered sequence of the supra- and infrahyoid activation pattern. These findings are consistent with those available in the literature. Koyama et al. found a significantly higher prevalence of inverted muscle activation patterns in dysphagic subjects, with the activation of infrahyoid muscles preceding the suprahyoid muscles [15]. Pre-onset muscle activation is a protective mechanism to prevent neuromuscular degeneration leading to kinematic and functional loss [51] and gives rise to prolonged swallowing times, which has been widely described in dysphagic subjects [52,53,54]. In fact, the appearance of swallowing with a pre-reflex phase of muscle activation was reported as a compensatory mechanism to adjust for age-related muscle weakness [51]. Koyama et al. found prolonged activation of infrahyoid muscles and shorter activity of suprahyoid muscles in dysphagic subjects, suggesting important changes in the timing of the initiation of swallowing-muscle activity [15]. This phenomenon may be caused by the forceful swallowing secondary to the lack of coordination of the swallowing muscles, which increases muscle activity amplitude [15]. Consequently, the previous activation of the infrahyoid muscle may be the origin of the significantly increased interaction strength in RIH→RSH, producing a loss in the physiological downward-dominant directional RSH→RIH in dysphagic subjects. The symmetry in the RSH⟷RIH interaction directionality may, thus, constitute a new dysphagia biomarker.

The different patterns found in RSH→RIH and LSH→LIH in dysphagic subjects may well be associated with the right hemispheric lateralization of the pharyngeal phase. This phenomenon was reported by other authors, who found a reduction in cortical swallowing-related activation in amyotrophic lateral sclerosis patients with progressive dysphagia in comparison to healthy controls, the right sensorimotor cortex being predominant [55]. This right hemispheric lateralization may be associated with the compensatory mechanism to coordinate the pharyngeal phase of swallowing thanks to brain plasticity [55]. The right hemispheric lateralization may also give rise to a delayed activation of LIH with respect to RIH, obtaining a significantly increased RIH→LIH interaction in dysphagic subjects, but not for LIH→RIH. The delayed triggering of the swallowing reflex for voluntarily initiated swallowing has also been observed in both amyotrophic lateral sclerosis patients and in dysphagia for suprabulbar palsy [56,57]. In the latter, when reflex swallowing could be triggered, it was slow and prolonged [57]. Our results suggest that the loss of symmetrical interaction of the bilateral infrahyoid muscles could be another dysphagia biomarker, which agrees with other authors who found that the bilateral muscle discoordination index was significantly greater in dysphagic than in healthy subjects [13]. Krasnodębska et al. also reported that patients with atypical swallowing patterns had significantly greater asymmetry of both the masseter and submental muscles [58].

Previous studies have shown that increasing the bolus consistency in healthy subjects prolonged the duration of oral and pharyngeal swallowing [59,60,61] as well as discrete and sequential swallowing [62]. Numerous studies have shown that highly viscous liquids significantly increase the duration of the supra- and some infra-hyoid muscle activations [51,59,63,64]. In comparison to swallowing saliva, the highest sEMG amplitude of the supra and infrahyoid muscles was obtained in healthy subjects swallowing 10 mL water and yoghurt [14], suggesting greater muscle recruitment. It was less safe for dysphagic patients to swallow thin liquids rather than thicker ones [65]. In this work, we also found that cG-causality was slightly higher for water and yoghurt than for saliva with no significant difference between them in healthy subjects, who seemed to have a good ability to fine-tune the activation pattern according to the type of bolus ingested, namely swallowing reserve [66], leading to similar interaction strengths among the boluses. The swallowing reserve decline due to neurological and neuromuscular diseases, muscle weakness caused by aging, and positional changes of swallowing-related organs [66]. This could explain the difference in the functional interactions among the boluses observed in dysphagic subjects. Generally, a decline in the swallowing reserve may cause a descent in the positions of the hyoid bone and larynx, a reduced antero-superior movement range for the hyoid bone and larynx elevation, larynx elevation delay, and a delay in the stimulation of the swallowing reflex [66]. The right hemispheric lateralization may also justify the preservation of the right-side muscle swallowing reserve while losing it in the left muscle group [55], which could explain the significant differences among the boluses in the LM→LSH and LSH→LIH interactions in dysphagic subjects.

### 4.3. Study Limitations

Despite its promising results, this study was not completely exempt from limitations. Firstly, significant differences in the interactions between the healthy and dysphagic subjects, as well as asymmetric interactions, were not consistent for all the boluses. Although the differences in bolus properties could yield inherently different swallowing responses, this finding may also be related to the limited sample size and the high intersubject variability due to population variance and intrasubject variability. The latter could have been affected by diverse biological factors, such as muscle fatigue, the volume of salivary secretions (which may vary according to the volume of liquid swallowed), the time interval between swallows, the number of trials, and the sequence of the food intake [46,67]. In addition, it was reported that dysphagic subjects may show significantly higher intrasubject variability between repetitions [68]. In this work, we only acquired sEMG data for a single swallow of each bolus. Repeated swallowing of these would provide a more robust characterization of the activation pattern of the muscles involved and would reduce intrasubject variability [46]. Future studies with repeated swallowing are still needed to corroborate our preliminary results. In addition, due to the limited sample size, we did not conduct the study by means of dysphagia aetiology to determine the difference in functional coordination between the subjects. Secondly, although cG-causality was originally formulated for linear stationary stochastic processes, sEMG has a nonstationary nature, which was the reason why we carried out the sliding window analysis. Finally, it should be noted that a multimodal analysis using electroencephalography and sEMG would provide a better understanding of the underlying electrophysiological mechanisms involved in swallowing, since the latter requires both voluntary and automatic control involving multiple brain regions [21].

## 5. Conclusions

In this work, we conducted a preliminary study on the utility of assessing the directed functional coordination between the masseter, supra-, and infrahyoid muscles during swallowing to detect possible alterations in functional coordination. We determined the physiological functional coordination pattern in normal swallowing (bilateral and ipsilateral supra- and infrahyoid-related activity) to be highly coupled. We also found a dominantly downward direction of the ipsilateral supra- and infrahyoid muscles in healthy subjects, which matches with the electrophysiology of swallowing, while the bilateral interactions were symmetric with no significant differences.

The main alterations in dysphagic subjects were found in the supra- and infrahyoid muscles, with no significant differences in the bilateral masseter and ipsilateral masseter↔suprahyoid muscle interactions. Specifically, we found that the right-to-left infrahyoid interaction was significantly higher in dysphagic subjects, suggesting that the loss in symmetry interaction of the bilateral infrahyoid could be potentially used as a dysphagia biomarker. We also found different pattern changes in dysphagic subjects in the left and right supra- and infrahyoid muscle interactions, with the right side being more resistant to a swallowing decline.

The loss in the asymmetric downward direction of the ipsilateral supra- and infrahyoid muscles could also be another dysphagia biomarker. Unlike healthy subjects, dysphagic subjects showed significant differences in the left masseter–suprahyoid, left suprahyoid–infrahyoid, left-to-right infrahyoid, and right infrahyoid–left masseter interactions, depending on the bolus consistency.

Our preliminary results suggested that the functional coordination analysis of swallowing muscles provided relevant information for evaluating motor control synergy and paved the way towards the identification of new, robust biomarkers for the early detection of dysphagia. Our method potentially contributed to developing a noninvasive and objective screening method for the early detection of swallowing dysfunction related to altered functional coordination that is not detectable by a videofluoroscopia swallowing study and could, therefore, be used to quantitatively assess the progress of dysphagia and the effectiveness of rehabilitation therapies.

## Figures and Tables

**Figure 1 sensors-22-04513-f001:**
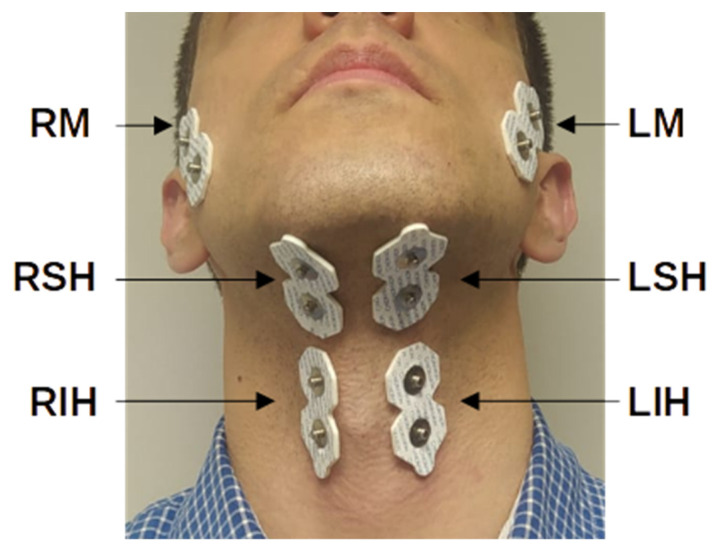
Electrode placement configuration for acquisition of multichannel sEMG signals. RM and LM: right and left masseter, respectively; RSH and LSH: right and left suprahyoid, respectively; RIH and LIH: right and left infrahyoid, respectively.

**Figure 2 sensors-22-04513-f002:**
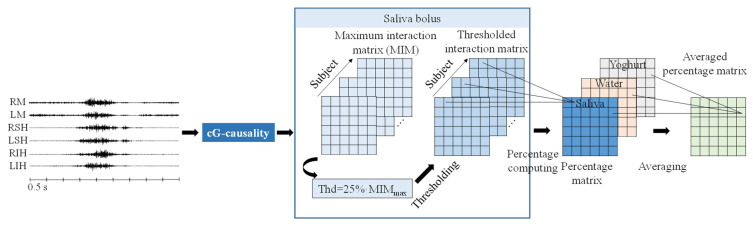
Flowchart of relevant interaction pairs with relatively high coupling strengths.

**Figure 3 sensors-22-04513-f003:**
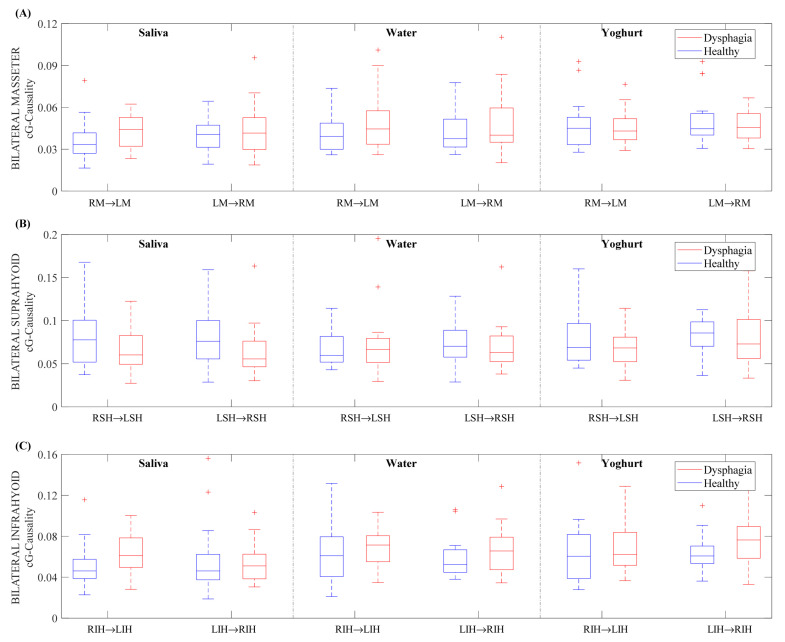
Box-and-whisker plot of the cG-causality values for bilateral muscles in healthy and dysphagic subjects for saliva, water, and yoghurt boluses. (**A**) Masseter, (**B**) suprahyoid, (**C**) infrahyoid.

**Figure 4 sensors-22-04513-f004:**
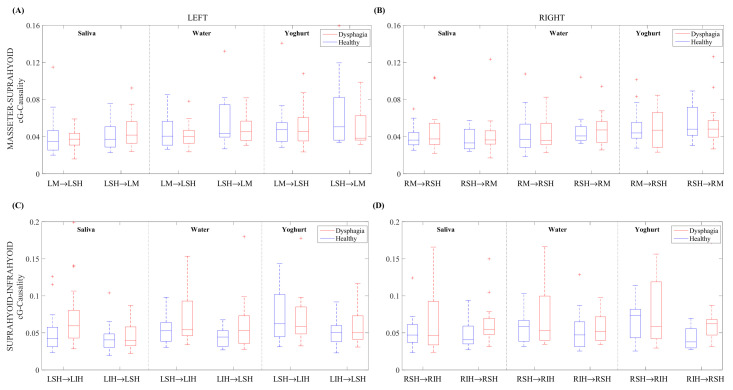
Box-and-whisker plot of the cG-causality for ipsilateral muscle interactions in healthy and dysphagic subjects for saliva, water, and yoghurt boluses. (**A**) LM⟷LSH, (**B**) LM⟷LSH, (**C**) LSH⟷LIH, and (**D**) RSH⟷RIH.

**Figure 5 sensors-22-04513-f005:**
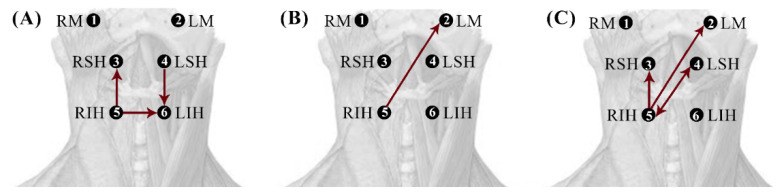
Schematic representations of statistically significant differences in the cG-causality of interaction pairs between healthy and dysphagic subjects for saliva (**A**), water (**B**), and yoghurt (**C**). Significantly higher and lower interactions in dysphagic subjects are shown in red and blue, respectively.

**Figure 6 sensors-22-04513-f006:**
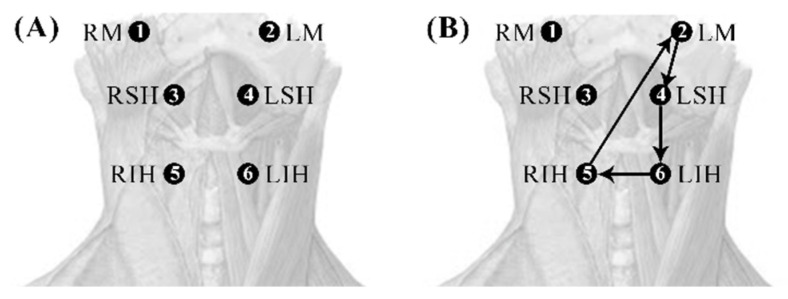
Diagrams with statistically significant differences in the cG-causality of interaction pairs between the different boluses in healthy (**A**) and dysphagic subjects (**B**).

**Figure 7 sensors-22-04513-f007:**
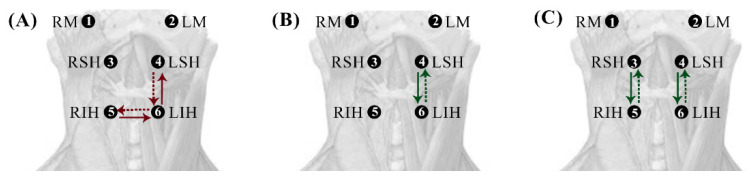
Diagrams with statistically significant differences in the cG-causality of the directionality of interaction pairs (asymmetry) for saliva (**A**), water (**B**), and yoghurt (**C**). Green and red arrows show healthy and dysphagic subjects, respectively. The continuous lines show the dominant direction.

**Table 1 sensors-22-04513-t001:** Average percentages of healthy/dysphagic subjects that exceeded the pre-established threshold for each interaction muscle pair. Bilateral, ipsilateral, and contralateral muscles are shaded in grey, green, and blue respectively. The overall average percentages of bilateral, ipsilateral, and contralateral muscle interactions are shown at the bottom of the table. Percentages above 85% are shown in bold.

		From					
		RM	LM	RSH	LSH	RIH	LIH
**To**	RM		78%|79%	**85%**|79%	66%|57%	84%|75%	65%|53%
	LM	72%|77%		73%|63%	**94%**|69%	59%|70%	**90%**|72%
	RSH	83%|68%	67%|63%		**100%|97%**	**88%|96%**	71%|58%
	LSH	67%|62%	76%|75%	**100%|91%**		69%|63%	**92%|85%**
	RIH	78%|63%	68%|65%	**96%|92%**	74%|75%		**98%|95%**
	LIH	57%|55%	82%|74%	82%|78%	**94%|95%**	**98%|97%**	
Average		Bilateral	**91%|89%**	Ipsilateral	**87%**|78%	Contralateral	68%|64%

## Data Availability

The data are not publicly available due to privacy or ethical restrictions.

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
