# Peer review of "Directed Functional Coordination Analysis of Swallowing Muscles in Healthy and Dysphagic Subjects by Surface Electromyography"

_sensors, 2022, doi:10.3390/s22124513_

Round 1

Reviewer 1 Report

In general, it is a quite nice manuscript. The authors provide a thorough characterization, which in most cases sufficiently supports their claims. However, there are still a few drawbacks in the manuscript that must be addressed. Some of the methods should be explained in more detail.

Line 149: The author's mention "The subjects’ skin was prepared to reduce skin-electrode impedance.". The details of the procedure should be added. 

Line 199: What was the basis of choosing a threshold at 25%? This should be explained. 

A major drawback of the paper is the lack of repetition of the presented results. While the authors do mention that as a limitation of the study (Line 421), confidence of inferring any conclusions becomes significantly low.

After careful evaluation of the manuscript, my recommendation is that the authors must perform minor revisions on the manuscript, prior to its publication.

Author Response

Reviewer 1

In general, it is a quite nice manuscript. The authors provide a thorough characterization, which in most cases sufficiently supports their claims. However, there are still a few drawbacks in the manuscript that must be addressed. Some of the methods should be explained in more detail.

Line 149: The author's mention "The subjects’ skin was prepared to reduce skin-electrode impedance.". The details of the procedure should be added. 

Authors: Thank you for your remarks. We are sorry for leaving out this important information in the previous manuscript. The skin preparation procedure is now described as follows:

“To reduce skin-electrode impedance, we first shaved the skin under the electrodes as required and then exfoliated with abrasion gel (Nuprep, Weaver and Company, Aurora, CO, USA) before cleaning the recording surface with isopropyl alcohol.”

Changes in the manuscript: This is included in lines 137 to 139 of the revised manuscript.

Line 199: What was the basis of choosing a threshold at 25%? This should be explained. 

Authors: Several methods have been proposed in the general literature to determine threshold [1], for example, absolute threshold, proportional threshold, consensus threshold and consistency threshold. None of these methods outperforms the others in all applications. Due to the great physiological variability between subjects, in this work we used the proportional threshold to remove the weakest interactions of each subject and then computed the percentage of subjects that exceeded this threshold for each interaction. Since our aim was to remove spurious interactions, we set the ad-hoc threshold to 25% of the maximum value of MIM (MIMmax).

To set the threshold, we took into account that the true ‘relevant’ interactions associated with swallowing should be reproducible in almost all healthy subjects, being present in almost 100% of the subjects, while some of these interactions may be altered in dysphagic subjects. We compared different threshold values ranging from 10-40% and obtained similar results for the 20-30% threshold range.

Higher threshold values reproduced the same phenomenon reported in the manuscript (bilateral interactions were ‘relevant’ in a greater percentage of subjects than ipsilateral interactions, which in turn were greater than contralateral interactions), but with lower population percentages than when using a value of 25% for the threshold. On the other hand, lower threshold values led to an overestimation of ‘relevant’ interactions and the inclusion of spurious ones, yielding very similar (and high) population percentages for bilateral, ipsilateral and contralateral interactions, which would limit the inference of physiological interpretations of the results. It also reduced the difference of the average percentage matrix between healthy and dysphagic subjects.

Changes in the manuscript: We have included this information from line 209 to line 215 of the revised manuscript.

“For the value of the threshold used, we aimed to achieve a trade-off between the sensitivity of detecting relevant interactions by a physiological interpretation and obtaining ‘spurious’ interactions. True interactions associated with swallowing should be reproducible in almost all healthy subjects, achieving approximately 100% of the subjects that exceeded this threshold. In addition, some of these interactions may be altered in dysphagic subjects. Specifically, in this work we compared different threshold values, ranging from 10-40%, and obtained similar results for threshold ranges of from 20 to 30%.”  

A major drawback of the paper is the lack of repetition of the presented results. While the authors do mention that as a limitation of the study (Line 421), confidence of inferring any conclusions becomes significantly low. After careful evaluation of the manuscript, my recommendation is that the authors must perform minor revisions on the manuscript, prior to its publication.

Authors: We agree with the reviewer. Indeed, we highlighted it as a limitation of the study in the original manuscript. We would like to point out that the lack of a consistently significant difference for the three boluses may be associated with the high inter-subject and intra-subject variability. The high inter-subject variability of swallowing due to population variance is well known. In addition, we should take into account the intra-subject variability between trials. Intra-subject variability could be affected by diverse biological factors such as muscle fatigue or the amount of the salivary secretion (which may vary according to the volume of liquid swallowed), the time interval between swallows, the number of trials and the sequence of the food intake [2][3]. Perlman et al. showed that the inter-subject temporal pattern agreement was 70% for saliva swallows and 45% for 10-ml liquid swallows in healthy subjects [4]. The high intra-individual pattern agreement in healthy subjects was 73% for saliva swallows and 65% for 10‑ml of swallows, which supports the concept of a central pattern generator [4]. Pack et al. analysed the reliability of supra- and infrahyoid EMG for evaluating swallowing in healthy subjects and suggested that the average of multiple swallows for each bolus is recommendable to obtain confident results [2].

In the literature, a larger variability has been reported between repetitions for dysphagic subjects. For instance, Archer et al. showed Duchenne muscular dystrophy patients with dysphagia showed significantly higher intra-subject variability of masseter peak amplitude [5]. This high intra-subject variability may mask their difference from healthy subjects to a certain degree and justify the lack of a consistently significant difference for the three boluses. In this regard, we believe that repeated swallowing would allow a more robust characterization of the activation pattern of the muscles involved and would reduce inter-subject variability, giving rise to more reliable results. However, repeated swallows could produce muscle fatigue, especially in the dysphagic population. 

Even so, in this work we analysed for the first time the directed functional coordination of swallowing muscles by cG‑causality and confirmed the alteration of functional coordination in dysphagic subjects. We determined the physiological functional coordination pattern in normal swallowing: high bilateral muscle coordination and ipsilateral supra- and infrahyoid interaction in a predominantly downward direction, which matches with the transit of bolus. We found that the appearance of a symmetric ipsilateral supra- and infrahyoid interaction and asymmetric bilateral infrahyoid interaction may constitute the hallmark characteristics of swallowing dysfunction. We believe that our results have a potential clinical value of general interest to the scientific-technical community. Also that our study paves the way to identifying new robust biomarkers for early detection of dysphagia or the quantitative evaluation of rehabilitation therapies.

Changes in the manuscript: We have included changes in the Discussion and Conclusions to clarify that the present work could be considered an exploratory study of the functional coordination analysis to assess swallowing dysfunctions, also that our results are preliminary (see lines 330, 364, 431, 440-442). Future studies are needed to corroborate these results. We have also included the inter-subject and intra-subject variability in Section 4.3 (lines 423 to 430).

Bibliography

[1]       C. R. Buchanan et al., “The effect of network thresholding and weighting on structural brain networks in the UK Biobank,” Neuroimage, vol. 211, no. December 2019, p. 116443, 2020.

[2]       M. W. Park et al., “Reliability of Suprahyoid and Infrahyoid Electromyographic Measurements during Swallowing in Healthy Subjects,” J. Korean Dysphagia Soc., vol. 11, no. 2, pp. 128–136, 2021.

[3]       S. M. Molfenter and C. M. Steele, “Physiological variability in the deglutition literature: Hyoid and laryngeal kinematics,” Dysphagia, vol. 26, no. 1, pp. 67–74, 2011.

[4]       A. L. Perlman, P. M. Palmer, T. M. McCulloch, and D. J. Vandaele, “Electromyographic activity from human laryngeal, pharyngeal, and submental muscles during swallowing,” J. Appl. Physiol., vol. 86, no. 5, pp. 1663–1669, 1999.

[5]       S. K. Archer, R. Garrod, N. Hart, and S. Miller, “Dysphagia in Duchenne muscular dystrophy assessed objectively by surface electromyography,” Dysphagia, vol. 28, no. 2, pp. 188–198, 2013.

Reviewer 2 Report

This work used the Conditional Granger Causality from surface electromyography 18 signals to analyze the directed functional coordination between different swallowing muscles in 19 both healthy and dysphagic subjects ingesting saliva, water and yogurt boluses.

The Data analysis section needs to be improved to include a detailed description of the algorithm. 

The significance of the results should be highlighted.

Comparison with the previous work is missing/insufficient.

Author Response

Reviewer 2

This work used the Conditional Granger Causality from surface electromyography signals to analyze the directed functional coordination between different swallowing muscles in both healthy and dysphagic subjects ingesting saliva, water and yogurt boluses.

The Data analysis section needs to be improved to include a detailed description of the algorithm.

­­­­­­Authors: Thank you for your comments. We omitted the description of the algorithm in the previous manuscript since, where we used the original algorithm formulated by Barnett [1]. Interested readers can easily find this information. Nevertheless, we agree with the reviewer that including this information could be helpful to the reader.

Changes in the manuscript: We have included the mathematical formulation of the conditional Granger causality in Section 2.2, as suggested by the reviewer (see lines 160 to 180 and 183 to 185 of the revised manuscript).

The significance of the results should be highlighted.

Authors. We are sorry for not sufficiently highlighting the significance of the results in the previous manuscript. For the first time this study analysed the directed functional coordination of swallowing muscles by means of cG-causality and confirmed the alteration of functional coordination in dysphagic subjects. We determined the physiological functional coordination pattern in normal swallowing: high bilateral coordination and ipsilateral supra- and infrahyoid interaction in a predominantly downward direction, which matches with the transit of the bolus. We also found that the appearance of a symmetric ipsilateral supra- and infrahyoid interaction and asymmetric bilateral infrahyoid interaction may constitute the hallmark characteristics of swallowing dysfunction. We are aware that a larger database both of the number of subjects and the number of swallows per subject would be recommendable to obtain more robust conclusions, and we believe that our method potentially contributes to developing a non-invasive and objective screening method for the early detection of swallowing dysfunctions related to altered functional coordination.  We believe that our results have a potential clinical value, are of general interest to the scientific-technical community and pave the way towards the identification of new robust biomarkers for the early detection of dysphagia or the quantitative evaluation of rehabilitation therapies.

Changes in the manuscript: We have expanded and rewritten the first and last paragraphs of the Conclusions.

Comparison with the previous work is missing/insufficient.

Authors: Thank you for your remark. As we mentioned in the previous manuscript, as far as we are aware, in this work we analysed for the first time the directed functional coordination of swallowing muscles by means of cG‑causality. Due to the novelty of the study, it was difficult to establish a direct comparison with previous studies. For this reason, we focused on the physiological interpretation of our results in the previous manuscript. Nevertheless, we agree with the reviewer that a more exhaustive comparison with previous works would be helpful to the reader.

Changes in the manuscript:

We have included the following text  the Discussion: from line 335 to 338, from line 343 to 349, from line 369 to 372, from line 396 to 397, from line 398 to 404, respectively, as follows:

“The high degree of supra and infrahyoid coupling agrees with other authors who found a significant positive correlation between these muscles [2]. We mostly found a high level of bilateral suprahyoid rather than bilateral infrahyoid coupling in healthy subjects, which agrees with our previous studies in assessing the functional coordination by cross correlation [3].”

“We also found a predominantly downward direction in ipsilateral supra- and infrahyoid muscles, which matches with the transit of the physiological descendent bolus. Under physiological conditions, the suprahyoid were activated 95 ms earlier than the infrahyoid muscles [4]. This may justify the relatively higher cG-causality value of suprahyoid-to-infrahyoid interaction (improved predictability of infrahyoid EMG activity with the known EMG activity from the suprahyoid muscle) than the infrahyoid-to-suprahyoid interaction.”

Pre-onset muscle activation is a protective mechanism to prevent neuromuscular degeneration leading to kinematic and functional loss [5] and gives rise to prolonged swallowing times, which has been widely described in dysphagic subjects [6][11][12].

“KrasnodÄ™bska et al. also reported that patients with atypical swallowing patterns had significantly greater asymmetry of both masseter and submental muscles [13].

Previous studies showed that increasing the bolus consistency in healthy subjects prolonged the duration of oral and pharyngeal swallowing [14][15][16] and discrete and sequential swallowing [17]. Numerous studies have shown that highly viscous liquids significantly increase the duration of the supra- and some infra-hyoid muscle activation [9][14][18][19]. In comparison to swallowing saliva, the highest sEMG amplitude of the supra and infrahyoid muscles was obtained in healthy subjects swallowing 10 ml water and yoghurt [3], suggesting greater muscle recruitment.  It is less safe for dysphagic patients to swallow thin than thicker liquids [20].”  

Bibliography

[1]       L. Barnett and A. K. Seth, “The MVGC multivariate Granger causality toolbox: A new approach to Granger-causal inference,” J. Neurosci. Methods, vol. 223, pp. 50–68, Feb. 2014.

[2]       H.-J. Lee and K.-D. Kim, “Analysis of the Correlation between Activity of the Suprahyoid Muscles, Infrahyoid Muscles and the New VFSS Scale in Stroke Patients with Dysphagia,” J. Korean Soc. Phys. Med., vol. 13, no. 4, pp. 19–25, 2018.

[3]       J. Garcia-Casado, G. Prats-Boluda, Y. Ye-Lin, S. Restrepo-Agudelo, E. Perez-Giraldo, and A. Orozco-Duque, “Evaluation of Swallowing Related Muscle Activity by Means of Concentric Ring Electrodes,” Sensors, vol. 20, no. 18, p. 5267, Sep. 2020.

[4]       M. W. Park et al., “Reliability of Suprahyoid and Infrahyoid Electromyographic Measurements during Swallowing in Healthy Subjects,” J. Korean Dysphagia Soc., vol. 11, no. 2, pp. 128–136, 2021.

[5]       J. Y. Ko, H. Kim, J. Jang, J. C. Lee, and J. S. Ryu, “Electromyographic activation patterns during swallowing in older adults,” Sci. Rep., vol. 11, no. 1, pp. 1–10, 2021.

[6]       P. Leslie, P. N. Carding, and J. A. Wilson, “Investigation and management of chronic dysphagia,” Br. Med. J., vol. 326, no. 7386, pp. 433–436, 2003.

[7]       M. P. Karnell and N. M. Rogus, “Comparison of Clinician Judgments and Measurements of Swallow Response Time,” J. Speech, Lang. Hear. Res., vol. 48, no. 6, pp. 1269–1279, Dec. 2005.

[8]       C. W. Lin, Y. C. Chang, W. S. Chen, K. Chang, H. Y. Chang, and T. G. Wang, “Prolonged Swallowing Time in Dysphagic Parkinsonism Patients With Aspiration Pneumonia,” Arch. Phys. Med. Rehabil., vol. 93, no. 11, pp. 2080–2084, Nov. 2012.

[9]       P. KrasnodÄ™bska, A. JarzyÅ„ska-Bućko, A. SzkieÅ‚kowska, and J. Bartosik, “Clinical and Electromyographic Assessment of Swallowing in Individuals with Functional Dysphonia Associated with Dysphagia Due to Muscle Tension or Atypical Swallowing,” Audiol. Res., vol. 11, no. 2, pp. 167–178, 2021.

[10]     H. Taniguchi, T. Tsukada, S. Ootaki, Y. Yamada, and M. Inoue, “Correspondence between food consistency and suprahyoid muscle activity, tongue pressure, and bolus transit times during the oropharyngeal phase of swallowing,” J. Appl. Physiol., vol. 105, no. 3, pp. 791–799, 2008.

[11]     W. V. Nascimento, R. A. Cassiani, C. M. Santos, and R. O. Dantas, “Effect of bolus volume and consistency on swallowing events duration in healthy subjects,” J. Neurogastroenterol. Motil., vol. 21, no. 1, pp. 78–82, 2015.

[12]     S. R. Youmans and J. A. G. Stierwalt, “Normal swallowing acoustics across age, gender, bolus viscosity, and bolus volume,” Dysphagia, vol. 26, no. 4, pp. 374–384, 2011.

[13]     L. J., S. E., S. C.M., and C. T., “Effects of liquid stimuli on dual-axis swallowing accelerometry signals in a healthy population,” Biomed. Eng. Online, vol. 9, p. 7, 2010.

[14]     T. Funami, S. Ishihara, M. Nakauma, K. Kohyama, and K. Nishinari, “Texture design for products using food hydrocolloids,” Food Hydrocoll., vol. 26, no. 2, pp. 412–420, 2012.

[15]     A. Igarashi et al., “Sensory and motor responses of normal young adults during swallowing of foods with different properties and volumes,” Dysphagia, vol. 25, no. 3, pp. 198–206, 2010.

[16]     D. Sifrim, N. Vilardell, and P. Clavé, “Oropharyngeal Dysphagia and Swallowing Dysfunction,” Front. Gastrointest. Res., vol. 33, pp. 1–13, 2014.

Reviewer 3 Report

This article details the results of electromyographic assessment in dysphagia. The article is interesting and well written. I recommend mentioning the functional aspects of dysphagia according to C.Kang and P.Krasnodebska e.g. as citation in line 95,96

Author Response

Reviewer 3.

This article details the results of electromyographic assessment in dysphagia. The article is interesting and well written. I recommend mentioning the functional aspects of dysphagia according to C.Kang and P.Krasnodebska e.g. as citation in line 95,96.

Authors: Thank you very much for your comments and suggestions.

Changes in the manuscript: We have mentioned the functional aspects of dysphagia in line 49 to 51 in the revised manuscript as follows:

“A videofluoroscopic swallowing study is the reference diagnostic method of assessing oropharyngeal dysphagia [1]. This technique involves patient exposure to ionizing radiation [1] and so is not recommended for patient follow-up when evaluating the effectiveness of rehabilitation, although it does not always identify neuromuscular abnormalities in pharyngeal or laryngeal physiology [1]. An example of this latter could be patients with muscle tension dysphagia, who present functional dysphagia but exhibit normal oropharyngeal and esophageal swallowing function as evidenced by videofluoroscopic swallow study [2][3]. Surface electromyography (sEMG) has emerged as a simple, non-radioactive […]”

Bibliography

[1]       M. Vaiman, “Standardization of surface electromyography utilized to evaluate patients with dysphagia,” Head Face Med., vol. 3, no. 1, pp. 1–7, 2007.

[2]       C. H. Kang, J. G. Hentz, and D. G. Lott, “Muscle Tension Dysphagia: Symptomology and Theoretical Framework,” Otolaryngol. Head. Neck Surg., vol. 155, no. 5, pp. 837–842, Nov. 2016.

[3]       P. KrasnodÄ™bska, A. JarzyÅ„ska-Bućko, A. SzkieÅ‚kowska, B. MiaÅ›kiewicz, and H. SkarżyÅ„ski, “Diagnosis in Muscle Tension Dysphagia,” Otolaryngol. Pol., vol. 74, no. 4, pp. 1–5, 2020.
